# Application of Machine Learning to Electroencephalography for the Diagnosis of Primary Progressive Aphasia: A Pilot Study

**DOI:** 10.3390/brainsci11101262

**Published:** 2021-09-24

**Authors:** Carlos Moral-Rubio, Paloma Balugo, Adela Fraile-Pereda, Vanesa Pytel, Lucía Fernández-Romero, Cristina Delgado-Alonso, Alfonso Delgado-Álvarez, Jorge Matias-Guiu, Jordi A. Matias-Guiu, José Luis Ayala

**Affiliations:** 1Department of Computer Arquitecture and Automation, Faculty of Informatics, Universidad Complutense de Madrid, 28040 Madrid, Spain; carlosmoralrubio@gmail.com (C.M.-R.); jayala@ucm.es (J.L.A.); 2Department of Neurophysiology, Institute of Neuroscience, Hospital Clínico San Carlos (IdISCC), Universidad Complutense de Madrid, 28040 Madrid, Spain; pbalugo@gmail.com (P.B.); adefraile@gmail.com (A.F.-P.); 3Department of Neurology, Institute of Neuroscience, Hospital Clínico San Carlos (IdISCC), Universidad Complutense de Madrid, 28040 Madrid, Spain; vanesa.pytel@gmail.com (V.P.); lucia_28028@hotmail.com (L.F.-R.); cristinadelgado1409@gmail.com (C.D.-A.); alfonso.delgado.alvarez@hotmail.com (A.D.-Á.); matiasguiu@gmail.com (J.M.-G.)

**Keywords:** electroencephalography, resting-state, primary progressive aphasia, biomarkers machine learning, K-Nearest Neighbors, frontotemporal dementia, Alzheimer’s disease, graph theory

## Abstract

**Background**. Primary progressive aphasia (PPA) is a neurodegenerative syndrome in which diagnosis is usually challenging. Biomarkers are needed for diagnosis and monitoring. In this study, we aimed to evaluate Electroencephalography (EEG) as a biomarker for the diagnosis of PPA. **Methods**. We conducted a cross-sectional study with 40 PPA patients categorized as non-fluent, semantic, and logopenic variants, and 20 controls. Resting-state EEG with 32 channels was acquired and preprocessed using several procedures (quantitative EEG, wavelet transformation, autoencoders, and graph theory analysis). Seven machine learning algorithms were evaluated (Decision Tree, Elastic Net, Support Vector Machines, Random Forest, K-Nearest Neighbors, Gaussian Naive Bayes, and Multinomial Naive Bayes). **Results**. Diagnostic capacity to distinguish between PPA and controls was high (accuracy 75%, F1-score 83% for kNN algorithm). The most important features in the classification were derived from network analysis based on graph theory. Conversely, discrimination between PPA variants was lower (Accuracy 58% and F1-score 60% for kNN). **Conclusions**. The application of ML to resting-state EEG may have a role in the diagnosis of PPA, especially in the differentiation from controls. Future studies with high-density EEG should explore the capacity to distinguish between PPA variants.

## 1. Introduction

Primary progressive aphasia (PPA) is a clinical syndrome secondary to the neurodegeneration of language brain regions and networks [1]. There are currently three main variants of PPA recognized in the literature: non-fluent (nfvPPA), semantic (svPPA), and logopenic variants (lvPPA) [2]. Diagnosis of PPA in the early stages is usually challenging. On the one hand, very mild word-finding difficulties may be present in aging, and the insidious onset of PPA symptoms limit an early identification [3]. On the other hand, there is a certain overlap between the PPA variants, especially between nfvPPA and lvPPA [4,5]. Neuropsychological batteries and language assessments are usually time consuming, and a high level of expertise is necessary for an adequate interpretation of the clinical findings. Although some novel and brief cognitive tests are being developed [6,7], neuroimaging and cerebrospinal fluid biomarkers are usually performed to confirm PPA diagnosis and the specific variant in each case. Structural magnetic resonance imaging (MRI) have shown adequate values of sensitivity and specificity for the diagnosis of svPPA, but diagnostic properties for the other variants are poorer [8,9]. Other more advanced MRI sequences show different patterns between PPA variants, but are not generally applicable routinely. Regarding positron emission tomography imaging, the 18F-FDG tracer has shown adequate values for diagnosis of the three variants of PPA, especially svPPA and lvPPA [10]. Amyloid tracers may distinguish between patients with amyloid deposition (generally associated with lvPPA) or not, but it does not have enough sensitivity to discriminate between subtypes of non-Alzheimer’s disease variants. Novel tracers, such as tau tracers, are still under investigation and are not usually available beyond research settings [11]. Consequently, the combination of several tools is often necessary to conduct an adequate diagnosis. However, some of these techniques are not available in all clinical settings, which jeopardizes the equality of opportunities. In recent years, there is an increasing interest in an accurate diagnosis of neurodegenerative disorders. Furthermore, early diagnosis may imply early access to language therapies, which have shown positive effects in PPA [12,13]. In addition, the classification of PPA into three clinical variants improves the prediction of the underlying pathology [14]. Thus, novel and cost-effective biomarkers are necessary for early detection and differential diagnosis between PPA variants.

One of the key processes in neurodegenerative disorders comprises the alterations in brain activity and network disruptions [15]. There are several methods for measuring brain activity, with differences in the spatiotemporal resolution and applicability. Some methods, such as single-unit recordings, have high spatial and temporal precision, but are invasive and are not applicable to large networks and clinical practice. Among the non-invasive methods, functional MRI, magnetoencephalography and electroencephalography (EEG) permit the assessment of brain activity across the entire brain [16]. These methods are generally well tolerated and applicable in clinical practice, and evaluate brain activity with a resolution on the scale of millimeters and centimeters. This means that each voxel of a conventional MRI or a channel of an EEG reflect the activity of thousands of neurons and billions of synapses [17]. In comparison to functional MRI, EEG shows lower spatial but higher temporal resolution. However, both techniques are regarded as useful for the assessment of brain activity and connectivity. As advanced computational algorithms promise to improve signal processing and filtering, noninvasive recording devices are increasingly being investigated and applied. Some approaches record neural potentials from the scalp, and depending on the intensity of recording, they can capture the activity of thousands of neurons. Multiple layers obstruct information transmission from the cerebral cortex to the scalp, resulting in signal amplitudes and spatial resolution that are reduced. The electrodes are also sensitive to external interferences such as eye movements, face movements, chewing, or swallowing, among others [18].

EEG is a widely available technique very useful for the diagnosis of epileptic disease. In the last years, quantitative EEG has also been confirmed as a helpful biomarker in the assessment of several neurodegenerative disorders [19]. These studies suggest a potential clinical application of EEG in the assessment of neurodegenerative disorders, either in the differential diagnosis between them or with other non-neurodegenerative causes, including psychiatric conditions [20]. Data regarding the application of EEG signal in PPA are scarce. In a recent study [21], three patients with nfvPPA and five with primary progressive apraxia of speech (two of them also showing aphasia) underwent EEG. A theta slowing was detected in almost all patients with nfvPPA, suggesting a potential clinical application. Another recent study has detected some particular findings in the analysis of EEG microstates in 8 patients with svPPA in comparison with controls and Alzheimer’s dementia [22].

Machine learning (ML) techniques may be helpful in improving the diagnostic performance of EEG, as has been shown in predicting epileptic seizures [23,24,25,26], Alzheimer’s disease [27], or depression [28,29]. The rationale for the application of ML to EEG is based on the following factors. First, the visual analysis of EEG is time-consuming and requires high levels of expertise. Second, changes in neurodegenerative disorders may be less visually evident than epileptiform activity, which also limits the inter-rater reliability. Third, filter settings, frequency bands and criteria for thresholds are not clearly defined in the setting of neurodegenerative disorders [30].

ML for EEG analysis may be divided into two approaches: feature-based and end-to-end. On the one hand, feature-based decoding algorithms have a long track record of effectiveness in various EEG decoding challenges [20,31]. The data are often represented by handcrafted and previously selected features in this approach. End-to-end decoding algorithms, on the other hand, allow raw or minimally pre-processed data as inputs [32,33]. To date, end-to-end deep learning has gotten much interest due to its success in other disciplines of research. At least for the extraction of the features, this technique might lead to better solutions or the discovery of unexpectedly informative characteristics, and it does not involve handcrafting. In terms of learning features, end-to-end models have a reputation for being “black boxes”.

In this study, we aimed to evaluate the potential of EEG as a biomarker for the diagnosis of PPA. For that purpose, the EEG raw signal was pre-processed in terms of feature transformations to enlarge the representation domain. We evaluated the diagnostic performance of EEG for the diagnosis of PPA, and the differential diagnosis between the three PPA Variants, applying ML models.

## 2. Materials and Methods

### 2.1. Participants

Forty patients with PPA were enrolled in this study. All patients met the current diagnostic criteria for PPA [1]. Patients were evaluated with a comprehensive language and neuropsychological protocol, which has been described elsewhere [34]. Structural MRI and FDG-PET were performed in all cases supporting the clinical diagnosis. Accordingly, patients were categorized as nfvPPA (*n* = 18), svPPA (*n* = 10), and lvPPA (*n* = 12). Twenty controls (control group, CG) were also included for comparison. Table 1 shows the details of the groups participating in the study.

The CG was obtained from patients that underwent EEG because of a previous history of syncope, but visual analysis of EEG, neuroimaging, and clinical follow-up were normal, excluding potential neurological disorders.

### 2.2. EEG Acquisition

EEGs were recorded in a resting state condition with the eyes closed and under the supervision of trained personnel. EEGs were acquired on a NicoletOne device of 32 channels, using the standard 10/20 system and referenced to A1. Time of acquisition was 20 min.

### 2.3. Preprocessing

Different signal transformations were applied, aiming to expand the amount of information. Signal preprocessing was performed following the pipeline implemented by [35] in EEGLAB Software (Matlab). These procedures try to minimize the external noise and artefacts that are usually present in a raw EEG signal. The following steps were conducted, which are also summarized in Figure 1:1.Time ranges selection. Original signals are too long to be analyzed and can contain some additional noise, so we manually selected those time ranges with higher quality in the signal representation to get the most accurate and clean signal. This process also considered the labels recorded during the EEG acquisition and clinical assessment, which notify about the state of the patient, unexpected events, or activities that could impact on the signal.2.High-pass filtering at 1 Hz. This filter was applied to remove baseline noise, remove noise introduced by sweating, and prepare the signal for ICA analysis.3.Apply CleanLine process with the following configuration: 10 Hz of bandwidth at 50 Hz line frequency. This preprocessing step removes line noise and related harmonics from each one of the scalp channels using a novel approach, as described in [36]. For that purpose, and for each sliding window over the original data, a multi-taper FFT is applied to transform the signal to the frequency domain; after that, the complex amplitude of the desired frequency is extracted. With that information, a noise signal in the frequency domain is generated and, finally, the time-domain associated noise signal that needs to be extracted from the original one is also created.4.Re-reference data to average. This is the most effective and easiest way to re-reference EEG data because it establishes that the summed up power across the scalp topography should sum zero. In other words, we removed the mean over all scalp channels to every single channel to make sure that all channels contribute with the same weight.5.Low pass filter at 40 Hz. This step was applied in order to remove any possible undesired high-frequency signal that was not removed by CleanLine. Although other investigations are looking for biomarkers in higher frequency ranges of EEG signal, most recent research works are focusing in lower frequency ranges [37]. For simplicity of our analysis and control of error sources, we have limited our work to the lower frequency bands.6.To apply ICA (Independent Component Analysis) to the signal. This method is a linear decomposition technique which aims to find the source signals from a set of mixed signals, as it occurs with EEG. Unlike PCA (Principal Component Analysis), ICA tries to retrieve those original signals that are maximally statistical independent in just one domain [38].7.To epoch data into windows of duration equal to one second without overlapping.8.Visual artifact rejection of epochs. As a final step, we reviewed manually all signals and all their epochs looking for artefacts or undesired signal events.

### 2.4. Quantitative EEG

QEEG, quantitative EEG, is the frequency domain transformation of the original EEG signal [39]. To obtain this transformed signal, the Discrete Fourier Transform method was applied over our sampled (at 500 Hz) EEG signal. Given a *x*(*n*) discrete EEG signal, the definition of Fourier Transform (FT) is as follows:(1)$$X_{k}=\sum_{n=0}^{N}x_{n}e^{-2\pi ikn/N},$$
for 0 ≤*k*≤N−1.

This transformation gave a transformed domain that increases the representation domains of the original signal and, hence, the information provided. We divided the total frequency range into non-overlapping frequency bands:Delta from 1–4 Hz.Ipsilon from 4–8 Hz.Alpha from 8–14 Hz.Beta from 14–30 Hz.Gamma from 30–45 Hz.OoB (out of bag) for frequencies higher than 45 Hz.

### 2.5. Wavelet Transformation

Wavelet Transform (WT) is the decomposition of the original signal into a set of basis functions consisting of contractions, expansions, and translations [40]. This is a similar approach to FT but using wavelet functions to achieve the transformed domain.

WT of the EEG signal was obtained by filtering repeatedly until reaching the desired level of decomposition. In each repetition, we applied a low-pass filter to obtain the approximation coefficient (CA) and a high-pass filter to obtain the detailed coefficient (CD). After every filtering stage, the signal was down-sampled by half the sampling frequency of the previous level.

A total of seven sequential subdivisions were applied until a sufficient number of transformations, all correctly subdivided in frequency, was achieved. This process provided eight signals, each one assigned to a different frequency range:Subband 1 from 125 to 250 Hz.Subband 2 from 62.5 to 125 Hz.Subband 3 from 31.2 62.5 Hz.Subband 4 from 15.6 to 31.2 Hz.Subband 5 from 7.8 to 15.6 Hz.Subband 6 from 3.9 to 7.8 Hz.Subband 7 from 1.9 to 3.9 Hz.Subband 8 from 0 to 1.9 Hz.

From all the extracted signals, sub-bands 1 and 2 were removed from the pipeline because neither of them offered any information after the application of step 5 in the preprocessing pipeline section (low-pass filter at 40 Hz).

### 2.6. AutoEncoders

Autoencoders are a specific type of neural networks architecture where the input is the same as the output. They compress the input into a lower-dimensional code and then reconstruct the output from this representation. The code is a compact “summary” or “compression” of the input, also called the latent-space representation. This novel modeling technique is also exploited in dimensionality reduction problems.

This architecture was created by using an Encoder-Decoder system (Figure 2). The first part, the Encoder, used fully-connected layers in which the number of input neurons is higher than the number of output neurons, to achieve that reduction of dimensionality. The second part, the Decoder, used fully-connected layers in which the number of input neurons is lower than the number of output neurons to achieve the reconstruction of the original signal.

In addition to this vanilla configuration, there are other complex configurations with multiple hidden layers for the encoding and decoding part, some of them even add noise to the input or to the intermediate part to force the neural network as a regularization technique to a better generalization.

As stated, this type of neural networks are usually applied as a technique for dimensionality reduction, but we decided to use them as a mechanism to transform the time domain (similarly to the FT and WT). We applied the Encoder to the EEG values along time, creating a new domain of features and information representation.

### 2.7. Graph Theory Analysis

Graph Theory Analysis (GTA) is a mathematical formalism used to model pairwise relations between objects. Here, we used the EEG sources from the different electrodes to generate the network that merges such data [41].

For that purpose, we generated a graph matrix, also called adjacency matrix, by calculating all pairs of partial correlations between all available channels or electrodes [42]. The absolute value operator was also applied to this matrix to achieve our final result, which in this case was an undirected weighted adjacency matrix.

Once the network was created, it was analyzed using different metrics:Node degree. This metric represents the number of links detected for every node.Path length. Mean of the shortest links present in the network.Clustering coefficient. Number of triangular connections in the network, divided by the theoretical maximum number of triangular connections. This variable represents the clustering capacity of the generated network.

In addition to these metrics, we also created a brain representation that shows each electrode as a node in a graph structure, and the connections that we obtained between those electrodes. An example of this representation ca be found in the Figure 3 and Figure 4 where the connections of the CG and PPA groups, filtered by alpha frequency range (from 8 to 14 Hz), are shown, respectively.

### 2.8. Data Analysis

#### 2.8.1. Binary Classification Model between PPA and CG

A classification model was generated to differentiate between CG and PPA patients based only on the extracted EEG features. Seven classification algorithms were evaluated: Decision Tree, Elastic Net (EN), Support Vector Machine (SVM), Random Forest (RF), k-Nearest Neighbors (kNN), Gaussian Naive Bayes (Gaussian NB) and Multinomial Naive Bayes (Multinomial NB).

The following pre-processing pipeline was applied to prepare the data for modelling:Train-test split. In this step we randomly generated train and test samples from the original dataset by applying 80% for training sample and 20% for test. This split was stratified, namely, the proportion of examples in each class is preserved into train and test samples.Scaling. We applied a MinMaxScaler method to each column in order to transform their range of values into the range [0, 1].Univariate Feature Selection. A feature selection step was applied to reduce the number of features to only 50 features from the original set (309). ANOVA F-value was computed for each column-target model and only the best 50 scores were selected.

In the training process, the selection of the best hyperparameters for each model was accomplished by a Bayes-search optimization algorithm (this optimization algorithm created a full space with all possible hyperparameter values and applied Bayes Theorem in order to find those exact values that minimized the error function). All models performed a binary classification using a 10-fold cross-validation using F1-Score metric (Equation 2) as the main metric. This metric was selected to optimize the classification problem, which was imbalanced. Precision (Equation 3), Sensitivity (Equation 4), Specificity (Equation 5) and Youden Index (Equation 6) are also displayed.
(2)$$F1-Score=\frac{2*Precision*Sensitivity}{Precision+Sensitivity}$$
(3)$$Precision=\frac{TruePositives}{TruePositives+FalsePositives}$$
(4)$$Sensitivity=\frac{TruePositives}{TruePositives+FalseNegatives},$$
(5)$$Specificity=\frac{TrueNegatives}{TrueNegatives+FalsePositives},$$
(6)YoudenIndex=Sensitivity+Specificity−1,

#### 2.8.2. Classification Model for All Groups

Following the same pipeline described in the binary classification model, a multiclass classification model was applied in order to distinguish between nfvPPA, svPPA, lvPPA and CG. The same pre-processing steps, hyperparameter tuning techniques and cross-validation options were applied here. All models performed a multiclass classification with 4 different classes (one per each group of patients) and using F1-Score metric as their main aim.

#### 2.8.3. Network Analysis

We transformed each EEG signal into a Network; this allowed to extract additional network metrics and enlarge the set of features per patient, but it also allowed to evaluate the differences between two brains in terms of activity according to these network metrics.

We also visualized the generated connections between EEG channels in each group of patients. This provided meaningful information about interactions of brain regions across the different groups of patients.

#### 2.8.4. Principal Components Analysis

In order to explain the complexity of our working dataset, we applied Principal Component Analysis (PCA) to reduce the dimensionality. PCA aims to find the directions of maximum variance in high-dimensional data and projects them onto a new subspace with equal or fewer dimensions than the original one. Hence, it reduces the number of features by combining them linearly. We performed a dimensionality reduction to only two principal components. Accordingly, the visualization of all subjects as data points is allowed by looking for the linear combination of all the extracted features into these two principal components. In this way, it is possible to visualize the multi-dimensional data distribution and evaluate how mixed are the data instances in the representation space.

## 3. Results

### 3.1. Classification Model between CG and PPA

Main metrics are summarized in Table 2.

Seven different models were evaluated for the binary classification: Decision Tree, EN, SVM, RF, kNN, Gaussian NB and Multinomial NB (Table 2). kNN model, a non-parametric supervised classification method, achieved the best performance, showing a Sensitivity of 0.88, an F1-Score value of 0.83 and a Specificity of 0.5. The confusion matrix from the best model (kNN) is shown in Figure 5, as well as its ROC curve in Figure 6.

The 20 most important variables used for training are depicted in Figure 7. A Decision Tree model is included in Figure 8. Regarding the most relevant variables, all variables except one were generated by the network transformations. Specifically, 40% from Node Degree, 50% from Clustering Coefficient, 5% Path Length, and 5% qEEG. Similarly, most features used in the Decision Tree algorithm are associated with network analysis.

### 3.2. Classification Model between All Groups

A multiclass model (4 classes) was developed to evaluate the possibility of automatic detection among all the PPA variants and CG. The same aforementioned models were evaluated (Table 3). Again, kNN model achieved the best performance. However, Sensitivity was 0.58 and F1-Score was 0.6. Confusion matrix for this model is shown in Figure 9.

As in the previous analysis, Figure 10 shows the graphical representation of the Decision Tree Model. In this case, decisions are mainly based on features obtained from network analysis and Autoencoder transformations.

## 4. Discussion

In this pilot study, we evaluated the diagnostic performance of a resting-state EEG obtained in clinical practice conditions for the diagnosis of PPA. We applied ML models, as they may be helpful to maximize the diagnostic capacity from many variables with no a priori hypotheses. In this regard, diagnostic performance was relatively high for the detection of patients with PPA in comparison with the control group. This suggests that there are certain EEG abnormalities that may be detected in patients with PPA. The most important features ranked by the algorithms for the classification and included in the decision trees algorithms involve mainly temporal and frontal channels in both hemispheres. Interestingly, features derived from network analysis obtained the best classification, emphasizing the role of graph theory in the analysis of EEG data [32]. These findings are consistent with recent investigations that are exploiting this new area of analysis [43,44].

Conversely, the application of EEG to the diagnosis of the specific variant of PPA did not achieve a satisfactory classification. Previous studies using quantitative data from EEG for the differential diagnosis of neurodegenerative disorders have obtained generally better results [31]. For instance, applying support vector machines, a 91% of accuracy was found to distinguish Alzheimer’s disease and dementia with Lewy bodies, and 88% for Alzheimer’s disease and Frontotemporal dementia [45]. Another study achieved a 93.3% of accuracy to classify between Alzheimer’s disease and Frontotemporal dementia [46]. However, these studies were performed with small samples, and were not replicated in larger studies [47]. The application of EEG to PPA is probably more challenging, due to the regional overlap between PPA variants in contrast to other disorders such as Alzheimer’s disease and Frontotemporal Dementia. In this regard, high-density EEG with a larger number of channels might obtain better results in the classification between PPA variants.

Our key insight is that machine learning itself can deal well with errors, qualitative and corrupted data and, more importantly for our purposes, integrate heterogeneous data from multiple domains. With this aim, our research work has enlarged the dataset to increase the information representation. The applied machine learning algorithms can jointly manage transformations from time to frequency domain, wavelet or network representation provided the setup parameters are carefully selected. In [48], a total of 49 experimental studies published from 2009 until 2020, which apply machine learning algorithms on resting-state EEG recordings from AD patients, were reviewed. These works did not evaluate the benefits of increased information representation in classification accuracy. Most of the studies focused on AD detection incorporating Support Vector Machines (SVM). Conversely, we found that classification algorithms based on distance (similar to kNN, where the function is only approximated locally and all computation is deferred until function evaluation) can improve performance.

The visualization of the multi-dimensional data used in our study, and the complexity of such dataset, has been performed with a PCA. Using this method, we found that patients are not clearly separated (Figure 11). This aspect is important for optimal performance in SVM models in the classification [49]). In contrast, kNN model, which is based on local distances, could lead to better predictions. This explains why we observed better performance of kNN model with respect to SVM model. Additionally, kNN model works better with a small number of features [50], which is our case after the application of the feature selection method. To follow this line, we replicated the same pipeline in the CG vs. PPA classification, skipping the feature selection phase. Thus, we compared the results of all models using only the best 50 features against using all generated features. As shown in Table 4, SVM model obtained better performance than kNN. However, the absolute values of the quality metrics (F1-score, precision, sensitivity and accuracy) explain the need for the feature selection process followed in our study.

The trend of increased information representation may be seen in recent works like [51] (still SVM in AD), or [52] (where, apart from EEG, a wide range of diagnostic tests were included). Our approach has focused exclusively in the classification possibilities of the EEG signal for the PPA, but we expect that our results could be improved by the addition of other tests such as neuroimaging, cognitive assessment, or genetics.

Patients included in this study fulfilled the current diagnostic criteria, with both MRI and FDG-PET supporting the diagnosis. As they were generally in early stages and EEG in comparison with controls was discriminative, these findings raise the possibility to explore in future studies the role of EEG in the clinical follow-up of patients with PPA, especially in the setting of clinical trials, in which reliable, reproducible and non-invasive endpoints are necessary [53].

Our study has some limitations. First, we included 40 patients, generally in early stages but not in the first consultation. Future studies should enroll a larger sample size and specifically focusing on patients in the early stages to confirm a potential role of EEG in the detection of PPA. Second, in this study, we only applied ML to EEG data. One of the main strengths of ML is the combination of multiple sources of information. Thus, the application of ML to studies including multimodality assessments (cognitive testing, MRI, PET) may be of interest to disentangle the best tools (isolated or in combination) for diagnosis of PPA [54].

## 5. Conclusions and Future Work

Our study shows that the application of ML to resting-state EEG may have a role in the diagnosis of PPA, especially in the differentiation from controls. EEG may have some advantages compared with other biomarkers.

ML techniques were applied to evaluate the possibility to automatically classify EEG data from PPA patients with respect to a control group. Our work showed that a feature expansion process can increase the information representation and achieve good classification accuracy, using mainly features from the graph-network representation of the EEG signal. The capability to classify PPA variants was also evaluated. Although lower, the classification capacity is still promising and advises further development of these automatic techniques for phenotype classification from EEG signals.

We are currently increasing the sample size to improve the classification accuracy of the models. In addition, we aim to enlarge the frequency range in the input dataset (over 45 Hz) to evaluate whether higher-frequency components may help the biomarker discovery with machine learning and deep learning methods.

## Figures and Tables

**Figure 1 brainsci-11-01262-f001:**
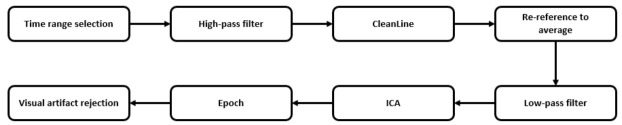
Preprocessing pipeline.

**Figure 2 brainsci-11-01262-f002:**
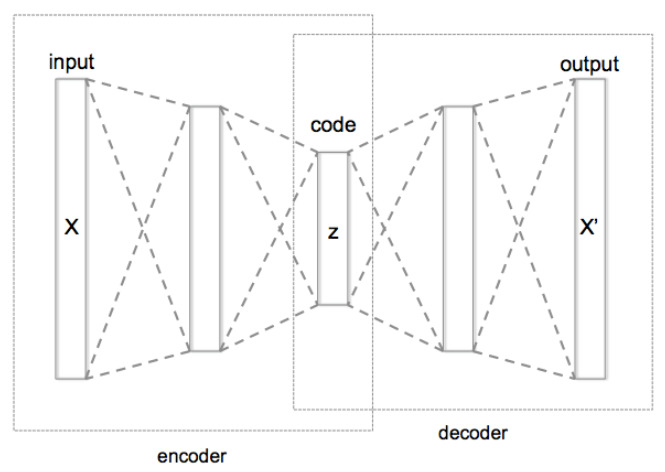
Encoder-Decoder architecture of AutoEncoder.

**Figure 3 brainsci-11-01262-f003:**
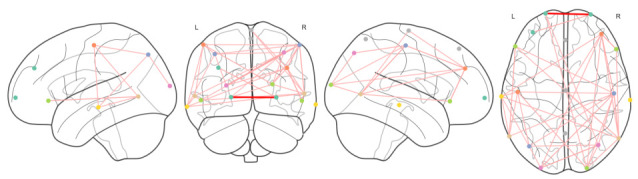
Network representation for CG group, including nodes, connections and their strength in alpha frequency band.

**Figure 4 brainsci-11-01262-f004:**
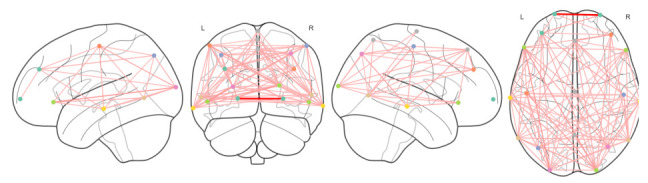
Network representation for PPA group, including nodes, connections and their strength in alpha frequency band.

**Figure 5 brainsci-11-01262-f005:**
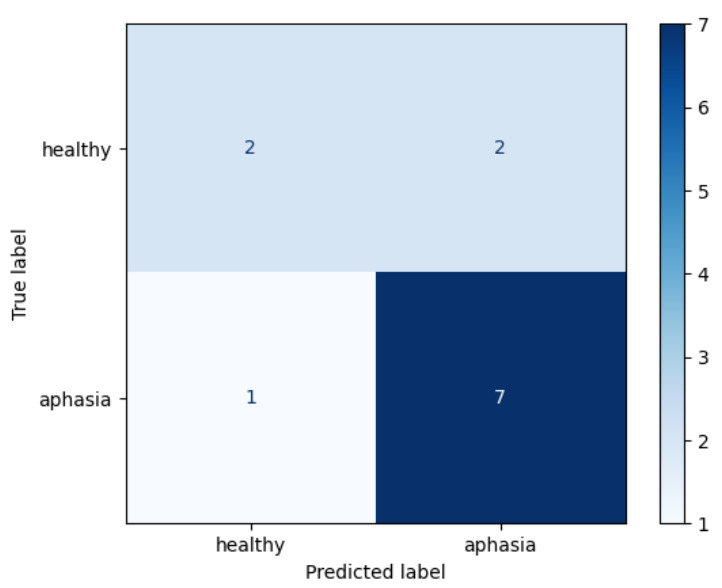
Confusion matrix from kNN binary classification (PPA vs. CG) model for the test set.

**Figure 6 brainsci-11-01262-f006:**
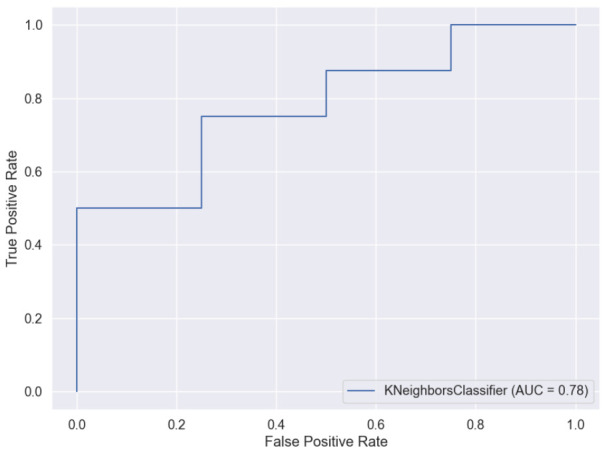
ROC curve from kNN binary classification (PPA vs. CG) model.

**Figure 7 brainsci-11-01262-f007:**
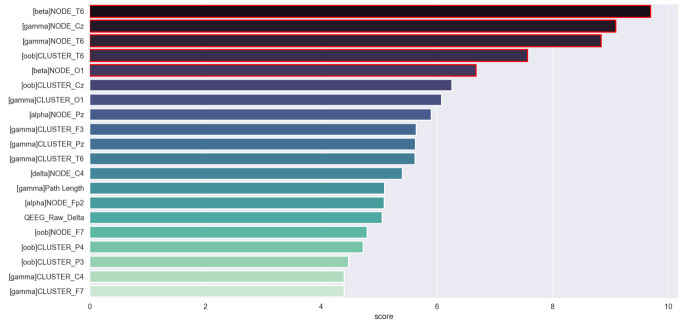
Representation of the 20 most important features.

**Figure 8 brainsci-11-01262-f008:**
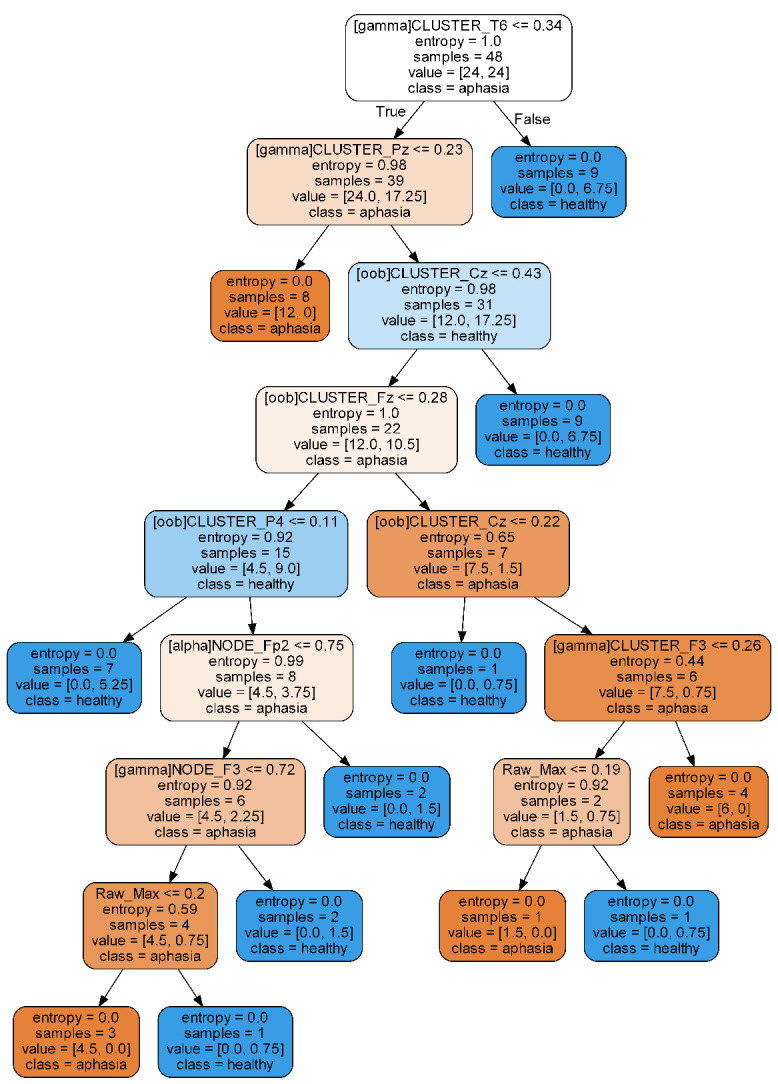
Representation of decisions in Decision Tree binary model. Scores represent the ANOVA F-value. Five most relevant variables are shown in red.

**Figure 9 brainsci-11-01262-f009:**
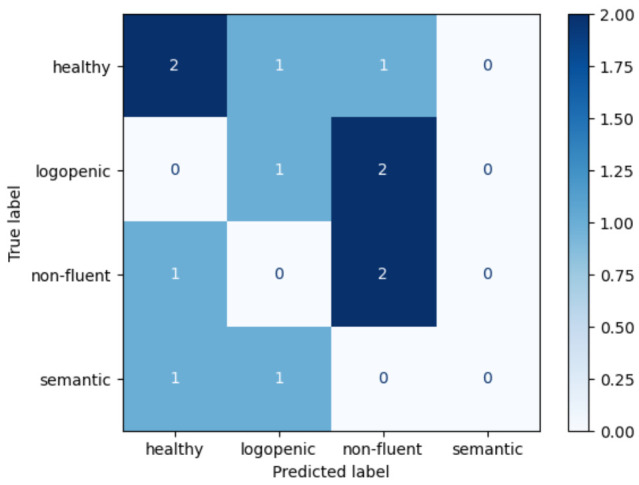
Confusion matrix from kNN multiclass classification model.

**Figure 10 brainsci-11-01262-f010:**
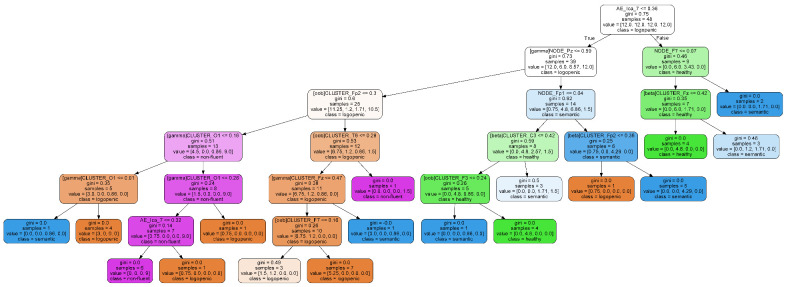
Representation of decisions in Decision Tree multiclass model.

**Figure 11 brainsci-11-01262-f011:**
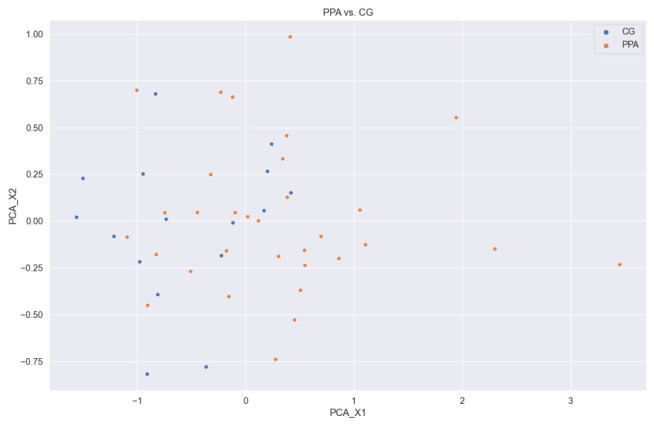
PPA vs. CG in PCA two dimensions.

**Table 1 brainsci-11-01262-t001:** Main clinical and demographic characteristics.

	PPA	nfvPPA	svPPA	lvPPA
**Number of participants**	40	18 (45%)	10 (25%)	12 (30%)
**Age**	68.7 ± 6.94	68.55 ± 7.29	66.80 ± 6.35	70.50 ± 6.97
**Women**	26 (65%)			
**Years of education**	13.90 ± 4.26	13.33 ± 4.41	14.20 ± 4.15	14.50 ± 4.35
**Years since symptom onset**	4.00 ± 2.25	4.83 ± 1.94	4.00 ± 2.98	2.75 ± 1.42
**ACE-III**	55.78 ± 26.59	71.76 ± 22.07	53.89 ± 15.72	48.00 ± 23.37
**CDR-FTLD (Sum of boxes)**	2.6 ± 1.81	2.22 ± 1.54	2.60 ± 1.67	3.16 ± 2.26

**Table 2 brainsci-11-01262-t002:** Metrics from classification models for PPA vs. CG.

Model	F1-Score	Precision	Sensitivity	Accuracy
**Decision Tree**	0.38	0.39	0.38	0.42
**kNN**	0.83	0.78	0.88	0.75
**SVM**	0.58	0.72	0.86	0.58
**Random Forest**	0.37	0.32	0.43	0.58
**Elastic Net**	0.4	0.33	0.5	0.66
**Gaussian NB**	0.78	0.9	0.75	0.83
**Multinomial NB**	0.73	0.73	0.75	0.75

**Table 3 brainsci-11-01262-t003:** Metrics from classification models for 4 groups (nfvPPA, svPPA, lvPPA, and CG).

Model	F1-Score	Precision	Sensitivity	Accuracy
**Decision Tree**	0.32	0.32	0.4	0.42
**kNN**	0.6	0.68	0.58	0.58
**SVM**	0.39	0.40	0.46	0.5
**Random Forest**	0.39	0.38	0.48	0.5
**Elastic Net**	0.34	0.31	0.33	0.38
**Gaussian NB**	0.27	0.25	0.31	0.33
**Multinomial NB**	0.2	0.2	0.25	0.25

**Table 4 brainsci-11-01262-t004:** Metrics from classification models for PPA vs. CG using all columns (with no feature selection).

Model	F1-Score	Precision	Sensitivity	Accuracy
**Decision Tree**	0.49	0.50	0.50	0.50
**kNN**	0.46	0.6	0.38	0.42
**SVM**	0.50	0.56	0.56	0.50
**Random Forest**	0.40	0.33	0.50	0.67
**Elastic Net**	0.40	0.33	0.50	0.67
**Gaussian NB**	0.37	0.32	0.44	0.58
**Multinomial NB**	0.56	0.56	0.56	0.58

## Data Availability

The data presented in this study are available on request from the corresponding author.

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
