# Peer review of "Application of Machine Learning to Electroencephalography for the Diagnosis of Primary Progressive Aphasia: A Pilot Study"

_brainsci, 2021, doi:10.3390/brainsci11101262_

Round 1

Reviewer 1 Report

This is a more than decent exploratory study. A bit of "throw it against the wall and see what sticks" but that comes with the territory and authors are upfront about it. 

Major issue:

The biggest issue of this manuscript is the inadequacy of discussion of obtained results. Some of these results are plain surprising (and, potentially, enlightening if looked into some more) like kNN consistently outperforming SVM (author's claim that it is to be expected for noisy datasets is extremely dubious to this reviewer and merits validation at least via references to any such studies not to mention that after all the manual preprocessing of the data in this study referring to it as "noisy" may be a stretch) or network/graph theory based features consistently outperforming the rest (again, there should be other studies that arrived at similar conclusions and at least attempted explaining them). There is a severe lack of insight into these surprising results as well as lack of any investigation as to why they turned out this way. Which is a pity because it could have been one of the main contributions of this manuscript. Luckily, it could be fixed rather easily.

Minor issues:

1. Every once in a while, there is an unfortunate sentence structure, e.g. "...it is insensitive to discriminate between..."

2. Claim of "it was demonstrated that useful ranges for EEG are under 40-45Hz" is extremely dubious. Authors mention epilepsy related EEG applications in line 48 so at the very least they should be aware of research related to high-frequency oscillations (in particular, ones recorded non-invasively) and their usefulness as a biomarker. 

3. Still, even after the application of a low pass filter at 40Hz there is still somehow a WT subband 1 signal corresponding to 125Hz to 250Hz being used among the features.

4. Please, add the frequency range to the captions of figures 3 and 4.

5. Speaking of figures, why are parentheses used in references to them like "are depicted in (Figure 7)" or "is included in (Figure 8)"?

6. Authors mention "sensitivity" and "recall" being synonyms but then use them in text interchangeably as opposed to being consistent and avoiding confusion.

Reviewer 2 Report

Authors conducted a cross-sectional study with 40 PPA patients categorized as non-fluent, semantic, and logopenic variants, and 20 controls. Resting-state EEG with 32 channels was acquired and preprocessed using several procedures. Undoubtedly, the application of ML to resting-state EEG may have a role in the diagnosis of PPA, especially in the differentiation from controls. 

My comments to the article are as follows:

- As part of Keywords, I propose to extend keywords with abbreviations used in Abstract.

- As part of the Introduction, I propose to extend the background with a broader reference to the methods of acquiring brain activity. For example, you can refer to: Data Acquisition Methods for Human Brain Activity, Analysis and classification of eeg signals for brain-computer interfaces, Book Series: Studies in Computational Intelligence, Springer from 2020.

- Fig. 8 and 10 is invisible, please scale it up - you can consider moving the schematic to the appendix.

- Conclusions should be distinguished in the article. Please post them after the Discussions section. Please also include your future plans for this research.

Round 2

Reviewer 2 Report

Dear Authors, 

Thank you for the responses and changes made to the article.

I have only one technical comment, in reference 18, you are missing the chapter title before the book name. The correct reference should read: Data Acquisition Methods for Human Brain Activity, Analysis and classification of eeg signals for brain-computer interfaces, Book Series: Studies in Computational Intelligence, Volume: 852, pp: 3-9, DOI: 10.1007 / 978-3 -030-30581-9_2, (2020)

In the other references, where there are a dozen Authors, it is particularly visible in the case of reference No. 14, I suggest listing several authors and then using the abbreviation "et al.".

Please also consider moving Fig. 10 as an attachment to the article.

Thank you for the changes made.